# DIFFUSION-BASED IMAGE-TO-IMAGE AUGMENTATION PRESERVING ACUPOINT LANDMARKS

## ABSTRACT

Deep learning models have achieved remarkable success in computer vision, yet their generalizability remains limited when applied to new domains or unseen tasks. Data augmentation is a common strategy to mitigate this issue, but traditional methods (e.g., rotation, scaling) often fail to capture the complex variations required by modern models. Recent advances in diffusion-based generative models, such as Stable Diffusion, enable high-quality image synthesis; however, their use in augmentation remains underexplored and faces challenges such as preserving biometric consistency and avoiding semantic drift. To address this issue, we propose a diffusion-based image-to-image augmentation workflow that transforms the original human images into new samples while keeping biometrical data unchanged, enriching the dataset without altering its key features. The resulted augmented dataset contains 225 synthetic anatomical models, each containing 44 images, resulting in a total of 9,900 images. Evaluation experiments demonstrate that the augmented dataset maintains 99.99

## 1 INTRODUCTION

Recent advances in machine learning have demonstrated remarkable capabilities across a wide range of tasks, including medical diagnostics, facial recognition, and human pose estimation. Traditional data augmentation methods, such as rotation, scaling, and color jittering, are simple to apply but are becoming more and more inadequate when facing modern machine learning tasks that require more diverse augmented data. These low-level transformations are not likely to introduce enough new data into the dataset. This limitation is especially critical in medical and biometric applications, where requires huge amount of data, but usually harder to acquire due to privacy issue. To address this challenge, diffusion models have recently emerged as a more powerful approach to data augmentation, which is able to produce realistic-level image with stable quality. Thus, a method based on diffusion models are developed to conduct an image-to-image transformation, re-generating the AcuSim dataset, which contains 63,936 synthetic RGB-D images with 174 annotated cervicocranial acupoints. By introducing the proposed data augmentation workflow, augmented dataset is able to be used directly by the following tasks, with no need to re-label the acupoints. The workflow is designed to preserve the structural features of the original models, especially the annotated acupoints, while introducing environmental factors such as lighting. Specifically, a controller program and two custom nodes are built to automate the process. Moreover, the generated dataset is evaluated in two approaches, a CNN-based model is used to evaluate the performance of the augmented dataset in a follow-up task. The network resulted in an almost perfect result with an accuracy of 0.99, showing that the augmented dataset can achieve the same level of performance as the original dataset. Another program is applied to detect facial landmark drift between the original and the augmented dataset. The analysis shows that most landmarks remain within a 5–8 pixel deviation, with only one point reaching about 10 pixels, which is still acceptable within clinical tolerance. These results demonstrate that the proposed augmentation method increases dataset diversity while maintaining characteristic consistency.

## 2 RELATED WORK

Data augmentation is central to improving robustness and performance when datasets are limited. Traditional methods (rotation, flipping, color jitter) are efficient but operate on low-level features

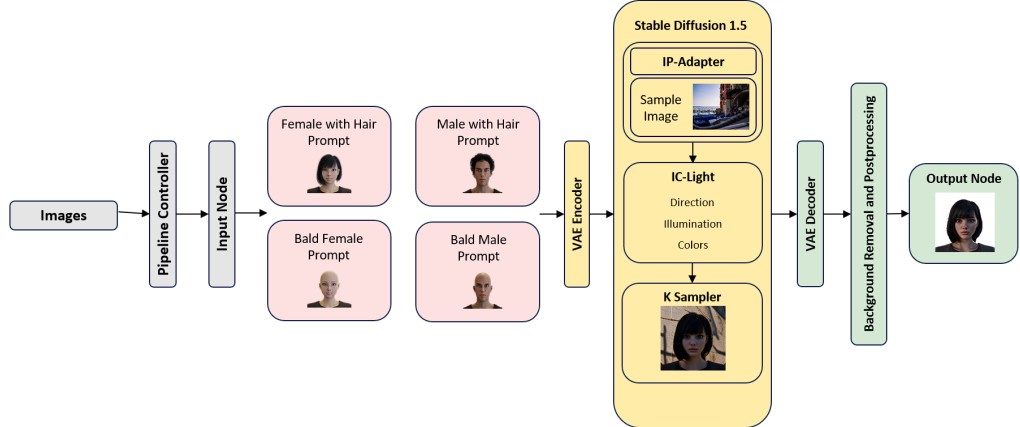

Figure 1: General Workflow

(Antoniou et al., 2018). To overcome this, model-based augmentation has been explored. For example, Loey et al. (2020) combined conditional GANs with classical augmentation for COVID-19 detection, and Choi et al. (2019) leveraged GAN-based augmentation for semantic segmentation in domain adaptation. However, GAN-based augmentation can be unstable and less controllable, which is problematic when strict structural consistency is required.

Diffusion models have advanced image synthesis, often surpassing GANs in fidelity and diversity (Dhariwal and Nichol, 2021). In medical imaging, diffusion approaches have been shown to generate high-quality images with improved diversity and stability relative to GANs (MedDiffusion, 2023). Concurrently, medical image analysis increasingly relies on synthetic and transferred knowledge: the AcuSim dataset (AcuSim, 2025) provides 63,936 synthetic RGB-D images with 174 annotated cervicocranial acupoints; transfer learning has enabled state-of-the-art results in data-scarce settings (e.g., malaria parasite detection) (ConvNeXt-TL, 2023). Outside medicine, diffusion-based augmentation has also been used to expand image datasets from reference images (EffDiffDA, 2023).

## 3 METHODOLOGY

### 3.1 OVERVIEW

Shown in Fig.1, we propose a diffusion-based image-to-image augmentation pipeline that preserves acupoint landmarks while transforming environmental factors (e.g., lighting, tone, background, hair colors) to strengthen dataset effectiveness. The workflow is automated and processes each image through a diffusion workflow. Real-world images are used as guidance in the background generation process to enhance re-lighting. The dataset used in this work is AcuSim (AcuSim, 2025), an open-source benchmark originally developed to automate acupuncture point annotation. It comprises 63,936 RGB-D images generated from 504 synthetic anatomical models, with 174 volumetric acupoints annotated per image. While this dataset provides a valuable resource for studying acupoint localization, prior work has primarily focused on synthetic models and did not explore transfer learning or validation on real human facial images, thereby limiting its applicability in real-world scenarios. Building upon this foundation, our goal is to expand AcuSim with diverse yet consistent facial variations while preserving anatomical landmarks, ensuring that the accompanying annotations remain directly applicable and improve generalization to real-life human acupoint annotation tasks. The workflow is built and tested in our development environment.

### 3.2 DIFFUSION WORKFLOW

The dataset includes labels for gender (M/F), body size (F/N/S), hairstyle, and eye state (O/C). These labels guide the reshaping process. The developed diffusion workflow combines text and image

guidance to control the overall tone and reshaping details. For text guidance, different prompts are used for male, female, bald male, and bald female samples. For image guidance, a single scene image is used to control the lighting vibe of the reshaped image. The lighting process is based on IC-Light to stabilize illumination and reduce changes around facial and head landmarks.

### 3.3 DIFFUSION PROCESS

The developed workflow uses a pre-trained latent diffusion model with S reverse steps. We splice the sample image from the original dataset into the process at $\lfloor St_0 \rfloor$, which controls how much the image is changed:

$$\mathbf{x}_{\lfloor St_0 \rfloor} = \sqrt{\tilde{\alpha}_{\lfloor St_0 \rfloor}} \, \mathbf{x}_0^{\text{ref}} + \sqrt{1 - \tilde{\alpha}_{\lfloor St_0 \rfloor}} \, \boldsymbol{\epsilon}, \quad \boldsymbol{\epsilon} \sim \mathcal{N}(\mathbf{0}, \mathbf{I}). \tag{1}$$

The model then runs step 0 with IP-Adapter weight $w_{ipa}$ for structure and IC-Light multiplier $w_{ic}$ for re-lighting. Other control variables include splice ratio $t_0$, CFG scale, sampler and scheduler, denoise strength, and the number of steps. In practice, a combination of VAE Encode, IP-Adapter, IC-Light, and a K-Sampler is used.

**VAE.** The variational autoencoder (VAE) is responsible for converting images between pixel and latent space. During encoding, a real frame (e.g., F_Brisen_F_HA_C_0141.jpg) is passed through VAE Encode, which compresses it into a latent representation. This representation is easier for the diffusion model to manipulate, while still retaining semantic information such as acupoint positions. After diffusion steps, decoding returns the latent to pixel space, producing a final augmented image. The VAE ensures all modifications occur in latent space, making reshaping faster and more robust than editing directly in pixel space.

**IP-Adapter.** The IP-Adapter is mainly for the image-to-image processes by providing visual controls. It will pass the characteristics of the input images to the generation processes, preserving information from the sample image.

**IC-Light.** IC-light is one of the core modules, interpreting lighting tune into the diffusion process. For instance, if a frame is captured with side lighting, IC-Light ensures that the augmented image still reflects this lighting condition rather than inventing a completely new one. This reduces unrealistic highlights or shadows that could otherwise confuse the model during training, while still allowing small variations in brightness or color temperature.

**K-Sampler.** K sampler controls the core sampling process in diffusion. It determines how random noise is gradually removed to generate the final image. The parameters in K sampler, for example, step, directly affecting the final image quality and fidelity. Moreover, the random seed controls the repeatability of the whole diffusion process. By changing the seed, the system can produce multiple distinct versions of the same base image.

## 4 IMPLEMENTATION

### 4.1 OVERALL DESCRIPTION

The workflow is divided into two main parts, controller and diffusion workflow. Due to the variety of the workflow, a python-based controller program is used to select different positive and negative prompt words for different sample IDs. The workflow is tested to be optimal when having dedicated prompt words for different set of samples. For example, a neutral-gender prompt is likely to blur the gender-different characteristics. Women samples are likely to have masculine characteristic when sharing prompt with male samples. Hair-related prompt with no bald-related-prompt-word may give hair to bald samples. According to (AcuSim, 2025)), the sample id contains model gender, name, body size, hairstyles, eye state, and image ID. We developed three dedicate prompt sets for female, male, and bold samples, and the controller program is used to detect the sample id to select different prompt set. After that, a custom-made input node will import a batch of images for processing, the quantity is depended on running environments. Similarly, after the imported images are processed, another custom-made output node will output the resulted images and restore the original dataset structures.

## 4.2 Parameters and Settings

The detailed settings of the proposed workflow are described in this section. The base model of the workflow is Stable Diffusion (SD) 1.5 with a built-in variational autoencoder (VAE). We chose SD 1.5 because it is one of the most widely used diffusion backbones, offering stable performance, wide community support, and relatively low computational cost compared to more recent versions. Moreover, it arguably performs the best compatibility with IC-Light, which is one of the most important modules of the workflow. The inclusion of VAE allows efficient encoding and decoding between pixel space and latent space, making the augmentation process both faster and more robust. For lighting control, we applied IC-Light with a multiplier of 0.3. IC-Light is a learning-based illumination correction model designed to enhance lighting uniformity across images. The 0.3 multiplier value was selected to maintain moderate lighting consistency. Strong artificial corrections for higher multiplier values may distort facial or head landmarks, while small value is likely to blur the human facial data by adding a strong lighting. The IP-Adapter module was used with a weight range of 0.3–0.6. The purpose of this setting is to preserve structural and identity-related features of the input image. In practice, it is mainly used to balance the mixing ratio of background and generated foreground image. An extreme low or high IP-Adapter weight is likely to result in a little influenced background or foreground image. The chosen range provides a balance, allowing the workflow to re-render the image with lighting changed, but leave no obvious trace of background reference image. This helps in the following background reduction and result generation module.

The sampling process was controlled by the K-Sampler, where we used the `dpmpp_2m` sampler with the `sgm_uniform` scheduler. The number of steps was set between 20 and 32, and the classifier-free guidance (CFG) scale was chosen within the range of 2.5-7. These values were selected to achieve a balance between generation speed, output quality, and fidelity. Fewer steps can reduce computational time but may lead to lower visual quality, while excessively large step counts increase cost without significant improvements. Similarly, the CFG scale controls the trade-off between prompt adherence and diversity; a moderate range ensures that the augmented images remain realistic without collapsing to identical outputs.

# 5 Evaluation and Results

## 5.1 Evaluation Methodology

**CNN-based acupoint localization and identification.** The overall network design follows the example network structure used in (AcuSim, 2025)). The network is used to perform an acupoint classification and coordinate regression task. Specifically, the dataset is divided into training and testing subsets, with an 80The model is trained for 200 epochs using the AdamW optimizer with an initial learning rate of 5e-5 and a weight decay of 1e-5. The loss function includes Euclidean distance root mean square error (RMSE) for the visibility test, cross-entropy loss for classification, and a soft-wing loss for coordinate regression.

**Facial-landmark-based difference comparison.** We assess geometric consistency between the original and augmented images using a facial-landmark-based pixel-offset analysis. MediaPipe is used to obtain selected 2D facial landmarks on each image in both datasets, before and after augmentation. Considering the stability of the landmark extraction process, we use eight landmarks on the human face: outer and inner canthi of both eyes (`LE_outer`, `RE_outer`, `LE_inner`, `RE_inner`), left and right mouth corners (`Mouth_L`, `Mouth_R`), Philtrum (upper-lip apex), and the midpoint of the nasal bridge between the two inner canthi (`NasalBridgeMid`). These points are widely used in facial analysis studies **?**. Thus, the evaluation settings form a robust facial-landmark difference-detection protocol.

Moreover, we add a discard mechanism because the dataset contains top-view and back-view samples that do not provide reliable landmarks for comparison. Specifically, a pair is discarded if: (i) landmark detection fails on either the original or the augmented image; (ii) the estimated face width, computed from the Face-Oval span, is below 80 pixels (typically extreme profile or back-of-head views); or (iii) fewer than 6 of the 8 selected keypoints are successfully detected in both images.

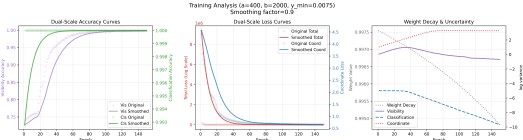

Figure 2: Enter Caption

For each retained pair and each keypoint $k$, we compute the Euclidean distance between the corresponding 2D locations:

$$d_k \;=\; \left\| \mathbf{p}_k^{(\mathrm{aug})} - \mathbf{p}_k^{(\mathrm{orig})} \right\|_2 . \tag{2}$$

$$\Delta x_k = x_k^{\mathrm{aug}} - x_k^{\mathrm{orig}}, \quad \Delta y_k = y_k^{\mathrm{aug}} - y_k^{\mathrm{orig}}, \quad d_k = \sqrt{(\Delta x_k)^2 + (\Delta y_k)^2}. \tag{3}$$

where (x,y) are pixel coordinates, aug stands for augmented dataset, and orig stands for original dataset.

## 5.2 RESULTS

**CNN evaluation.** Shown in Fig.2, for the CNN evaluation, the overall loss decreased steadily, and the visibility accuracy improved from 0.73 in the first epoch to a stable level close to 0.9 in the later epochs. The classification accuracy quickly spiked to 0.99, and went steady along the process, indicating the network is able to consistently identify the correct acupoint. Meanwhile, the coordinate regression loss decreased from 4.56 in the first epoch to values between 1.0 and 2.0 after convergence. These results demonstrate that the augmented dataset supports robust model training without loss of performance compared to the original dataset. For the facial recognition evaluation, most keypoints showed average displacements in the range of 5–8 pixels, including the outer and inner canthi, the mouth corners, and the nasal bridge midpoint. The philtrum point showed a slightly larger offset at around 10.1 pixels. The philtrum is a relatively unusual landmark, which is likely to be harder to locate. According to (Anthropometric Study of Philtrum (Face) and other nasal parameters in Nepal), philtrum is classified as soft tissue, which may vary with small facial expression changes. But still, 10.1 pixels are regarded within the tolerance of 5mm according to conversion method mentioned in (AcuSim, 2025)).

**Facial-landmark evaluation.** For the CNN evaluation, the overall loss decreased steadily, and the visibility accuracy improved from 0.73 in the first epoch to a stable level close to 0.9 in the later epochs. The classification accuracy quickly spiked to 0.99, and went steady along the process, indicating the network is able to consistently identify the correct acupoint. Meanwhile, the coordinate regression loss decreased from 4.56 in the first epoch to values between 1.0 and 2.0 after convergence. These results demonstrate that the augmented dataset supports robust model training without loss of performance compared to the original dataset. For the facial recognition evaluation, most keypoints showed average displacements in the range of 5–8 pixels, including the outer and inner canthi, the mouth corners, and the nasal bridge midpoint. The philtrum point showed a slightly larger offset at around 10.1 pixels. The philtrum is a relatively unusual landmark, which is likely to be harder to locate. According to (Anthropometric Study of Philtrum (Face) and other nasal parameters in Nepal), philtrum is classified as soft tissue, which may vary with small facial expression changes. But still, 10.1 pixels are regarded within the tolerance of 5mm according to conversion method mentioned in (Acusim).

## 6 CONCLUSION

In this work, we proposed a diffusion-based image-to-image augmentation workflow for human facial data augmentation. The workflow was designed with a controller program, as well as a custom input and output node, enabling automatic process for samples with different characteristics. Evaluation results demonstrated that the augmented dataset maintained high performance in a CNN based acupoint localization and classification task. A MediaPipe based pixel difference comparison is also

conducted, with most landmarks falls in between 5-8 pixels, and only one landmark shows a higher difference of 10 pixels. These findings indicate that our method successfully increases data diversity while preserving key structural and biometric information. Future work may explore extending this workflow to other anatomical regions and integrating more advanced techniques to further improve efficiency and generalizability.

ACKNOWLEDGMENTS

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

## A  APPENDIX

Additional details and qualitative examples can be included here.

