# OpenReview forum: "A Diffusion-Based Data Augmentation Approach for Synthetic Human Portraits Dataset"
_ICLR.cc/2026/Conference — Submitted to ICLR 2026_

### Official Review · Reviewer_5i3i · 2025-10-15

**Soundness:** 1
**Presentation:** 1
**Contribution:** 1
**Rating:** 0
**Confidence:** 5

**Summary:**

The paper proposes a diffusion-based image-to-image augmentation method that attempts to preserve facial/acupoint landmarks. The pipeline uses a combination of Stable Diffusion (SD-1.5), an IP-Adapter, an IC-Light module, and a choice of noise level injection to control how much the original image is retained. The motivation is to generate new visual diversity (background, lighting, skin tone, hair) while keeping acupoint labels valid, so that one can reuse existing labels without relabeling. The authors build a new “AcuSim” dataset (225 subjects × 44 views ≈ 9,900 images) and report very high accuracy (~99 %) on a CNN for acupoint classification and small landmark drift (5–8 pixels for most points).

**Strengths:**

1) The goal is compelling: reduce labeling burden by generating new but label-consistent data.
2) The idea of mixing partial guidance (injection of original image at noise level) to maintain structure is promising.

**Weaknesses:**

1)  The paper is not ready for submission for instance in pp 5 it mentioned that "Figure 2: Enter Caption". As we can see in pp 2 in "Figure 1: General Workflow"
2) The method is largely an engineering combination of existing modules (Stable Diffusion, IP-Adapter, lighting control). The novelty lies in their orchestration for this task, but that might be considered incremental.
3) The “99 % accuracy” result is too good to be believable without more context: which baseline, which dataset split, how many classes, variance across runs, and is the model overfitting?
4) There is no test of generalization: i.e. train on augmented data, test on new subjects or different lighting/pose settings. Does the augmentation help or hurt out-of-distribution generalization?

Missing relevant references in literature review

1) Context-guided Responsible Data Augmentation with Diffusion Models

2) Effective Data Augmentation With Diffusion Models

3) Diffusion models: A comprehensive survey of methods and applications

4) GenMix: Effective Data Augmentation with Generative Diffusion Model Image Editing

5) DiffuseMix: Label-Preserving Data Augmentation with Diffusion Models

**Questions:**

1) Compare the proposed method vs. classical augmentation (flip, color jitter, random crop) and, ideally, a GAN-based or identity-preserving diffusion baseline on the acupoint task.
2) Provide ablation experiments: how much each module (IP-Adapter, IC-Light, injection strength) contributes, and report performance vs drift tradeoffs across settings.
3) Test generalization: train with augmented + original, evaluate on unseen subjects, unseen lighting, or even a held-out real dataset to ensure no overfitting.
4) If possible, show that downstream tasks beyond acupoint classification (e.g. landmark regression, localization) benefit from the augmentation.

---

### Official Review · Reviewer_xTmF · 2025-10-26

**Soundness:** 1
**Presentation:** 1
**Contribution:** 1
**Rating:** 0
**Confidence:** 5

**Summary:**

This paper is an incomplete submission.

**Strengths:**

N.A.

**Weaknesses:**

This paper is an incomplete submission.

**Questions:**

N.A.

---

### Official Review · Reviewer_nwNu · 2025-10-30

**Soundness:** 1
**Presentation:** 1
**Contribution:** 1
**Rating:** 0
**Confidence:** 1

**Summary:**

The paper appears to be incomplete

**Strengths:**

The paper appears to be incomplete

**Weaknesses:**

The paper appears to be incomplete

**Questions:**

The paper appears to be incomplete

**Details Of Ethics Concerns:**

The paper appears to be incomplete

---

### Meta-Review · Area_Chair_2DoK · 2026-01-07

**Summary:**

Incomplete submission; not conference-ready.

**Reviewer Concerns:**

Submission is incomplete. No rebuttal provided to address concerns.

**Reviewer Scores:**

nwNu: 0

xTmF: 0

5i3i: 0

---

### Decision · Program_Chairs · 2026-01-26

Reject